# A Serodiagnostic IgM ELISA to Detect Acute Cytauxzoonosis

**DOI:** 10.3390/pathogens11101183

**Published:** 2022-10-14

**Authors:** Yun-Fan Kao, Rebecca Spainhour, Shannon R. Cowan, Laura Nafe, Adam Birkenheuer, Mason V. Reichard, Craig A. Miller

**Affiliations:** 1Department of Veterinary Pathobiology, College of Veterinary Medicine, Oklahoma State University, Stillwater, OK 74075, USA; 2Department of Veterinary Medicine and Surgery, College of Veterinary Medicine, University of Missouri, Columbia, MO 65211, USA; 3Department of Clinical Sciences, North Carolina State University, Raleigh, NC 27606, USA

**Keywords:** cytauxzoonosis, *Cytauxzoon felis*, domestic cats, diagnostic assay, ELISA, serology

## Abstract

Cytauxzoonosis is a tick-borne infectious disease affecting domestic cats with high mortality and limited treatment modalities. Because early diagnosis and therapeutic intervention are crucial to survival of infected cats, the objective of this study was to develop an ELISA capable of detecting cytauxzoonosis and differentiating acute vs. chronic infection in clinical feline blood samples. A microsphere immunoassay (MIA) was developed to evaluate the production of *Cytauxzoon felis*-specific IgM and IgG antibodies in serial plasma samples from cats with experimental *C. felis* infection by targeting a *C. felis*-specific transmembrane protein (c88). Recombinant c88 protein was utilized to develop indirect ELISAs to detect IgM and IgG antibodies in clinical plasma samples from: PCR-positive cats with acute *C. felis* infection (*n =* 36), *C. felis*-negative cats with pyrexia (*n* = 10), healthy *C. felis*-negative cats (*n =* 22), and chronic *C. felis* carriers (*n = 4*). Anti-c88 IgM antibodies were detectable at day 12 post-tick infestation in cats with experimental *C. felis* infection (within 24 hours of developing clinical signs), while anti-c88 IgG was detectable at day 15 post-tick infestation – indicating IgM could be used to detect early infection. Using a cut-off value of 19.85 percent positive, the *C. felis* IgM ELISA detected acute cytauxzoonosis in 94.44% (34/36) of cats presented with clinical signs of acute cytauxzoonosis with 100% specificity (indicating a “Strong Positive” result). When a lower cutoff of 8.60 percent positive was used, cytauxzoonosis was detected in the 2 remaining PCR-positive cats with 87.88% specificity (indicating of a “Weak Positive” result). One *C. felis*-negative, febrile cat had high IgG, and chronic carriers had variable IgM and IgG results. Combined interpretation of IgM and IgG ELISAs did not reliably differentiate acute vs. chronic infection. While further validation on assay performance is needed, the *C. felis* IgM ELISA is a promising test to detect acute cytauxzoonosis and can be utilized to develop a point-of-care test for clinical use.

## 1. Introduction

Cytauxzoonosis is a fatal disease of domestic cats with limited therapeutic modalities and a narrow timeline for instituting treatments. During acute infection, cats develop severe clinical illness with rapid disease progression associated with the schizogenous phase of parasite replication, in which large schizont-laden monocytes occlude vascular lumina in vital organs [1,2]. In the southcentral and southeastern United States, fatal cases of cytauxzoonosis in domestic cats are common and mortality in these regions can reach 100% without treatment [3]. Unfortunately, there is no vaccine to prevent *C. felis* infection, and attempts to develop effective anti-protozoal therapies have been met with limited success [4,5,6,7]. Current therapies (atovaquone + azithromycin) have been shown to increase survival up to 60% in cats with cytauxzoonosis, but are hindered by a narrow window in which treatment must be instituted early in the course of disease in order to be effective [7,8,9]. Because of the rapid disease progression and severe clinical consequence, early detection and diagnosis of cytauxzoonosis is crucial for effective treatment.

Although numerous diagnostic methods are available to aid veterinarians in diagnosing cytauxzoonosis, each technique has its limitations. Currently, diagnosis of cytauxzoonosis relies on cytology and confirmatory molecular testing such as quantitative polymerase chain reaction (qPCR). Cytologic diagnosis of cytauxzoonosis is achieved by identifying intra-erythrocytic piroplasms and/or schizont-laden macrophages in blood smears from infected cats, and although blood smear evaluation is cost efficient and can be performed in a clinical setting, normal structures like Howell-Jolly bodies or water artifacts resulting from poor or inadequate slide preparation can be mistaken as piroplasms by underexperienced individuals. In addition, schizont-laden macrophages are not always found in routine blood smears, and because piroplasms (merozoites) may not be present in erythrocytes until late stages of the parasite life cycle in cats, diagnosis of early *C. felis* infection may be missed [2,7,8]. Schizont-laden macrophages can also be identified in tissue aspirates from peripheral lymph nodes, spleen, liver and the lungs, but there are inherent risks and difficulties in acquiring these samples for analysis, and the sparse availability of ultrasound equipment might hinder veterinarians from utilizing this diagnostic method. [7,8]. Alternatively, blood samples from cats with suspected infection may be submitted to diagnostic laboratories that offer real-time PCR for the detection of *C. felis* DNA. Although considered a “gold-standard” in diagnosing active *C. felis* infection, real-time PCR and other available PCR assays can be time- and cost-prohibitive, and PCR is not readily available in the clinical setting; highlighting a critical need for a rapid, clinical test for definitive diagnosis in order to improve therapeutic outcomes.

Enzyme-linked immunosorbent assay (ELISA) is a plate-based assay technique designed for the detection and quantification of soluble proteins or peptides and is widely used in the biomedical field to detect various substances of interests including antigens from pathogens, antibodies, and hormones. The principle of ELISA involves the high affinity of antigen-antibody interaction and is accomplished by immobilizing a target of interest to a solid surface (microplate), which is then complexed to an antibody linked to a reporter enzyme. Detection is achieved by measuring the activity of the reporter enzyme via incubation with appropriate substrate to produce a measurable product. Many commonly used bedside rapid diagnostic test kits in veterinary clinics are designed based on ELISA methodology such as the SNAP^®^ 4DX Test (IDEXX Laboratories, Inc. Westbrook, Maine, USA) for common canine vector-borne diseases, SNAP^®^ FIV/FeLV Combo Test (IDEXX Laboratories, Inc. Westbrook, Maine, USA) for feline retroviruses, WITNESS^®^ Lepto Rapid Test (Zoetis LLC, Kalamazoo, MI, USA) for leptospirosis and the Cardiopet^®^ proBNP Test (IDEXX Laboratories, Inc. Westbrook, Maine, USA) for detection of preclinical cardiac diseases. These rapid tests have provided tremendous advantages to the veterinary practitioners in disease screening and implementation of appropriate treatment strategies for veterinary patients in a timely fashion. 

IgM antibodies are the first to be elicited in an immune response to infection and play an important role in generating early immunity, whereas IgG antibodies are produced at later stages and associated with prolonged immunity [10]. Thus, the objectives of this study were to (1) evaluate the dynamics of the anti-*C. felis* IgM and IgG antibody response in cats with acute cytauxzoonosis and (2) use this to develop an ELISA assay to detect acute cytauxzoonosis in clinical samples from domestic cats. Recent studies in *C. felis* genome sequencing and subsequent protein microarrays have identified differentially reactive *C. felis* antigens that are expressed during the early stages of the parasite life cycle [11], highlighting several potential targets to detect antibodies generated in response to acute *C. felis* infection. In the current study, we demonstrate that although cats generate both significant IgM and IgG antibody responses against *C. felis* during acute infection, anti-*C. felis* IgM antibodies are generated earlier in the course of infection than IgG, and detection of anti-*C. felis* IgM coincides with the onset of clinical signs. An indirect ELISA targeting IgM antibodies to a reactive *C. felis* transmembrane antigen was developed and results in accurate detection of *C. felis* infection in 94.44% of cats with clinical signs of acute cytauxzoonosis at 100% specificity—providing a relatively non-invasive assay with the potential to be developed into a rapid point-of-care test for clinical use.

## 2. Materials and Methods

### 2.1. Control Samples

Anticoagulated (EDTA) plasma sample from a cat presented to Oklahoma State University (OSU-VMH) with signs of acute cytauxzoonosis was utilized as positive control plasma for the IgM ELISA. EDTA-anticoagulated plasma sample from a cat at 19 days after experimental *C. felis* infection [12] was utilized as positive control plasma for the IgG ELISA. Cytauxzoonosis was confirmed both cytologically and via droplet digital PCR (ddPCR) [12]. EDTA-anticoagulated plasma samples from 18 specific pathogen-free cats housed at Oklahoma State University Animal Resources and from another *n =* 4 specific pathogen-free cats from another study [12] were available as negative control samples. Negative control samples (*n =* 22) were classified into group 1.

### 2.2. Experimental Infection Samples

Anticoagulated (EDTA) plasma samples from specific pathogen-free cats (*n =* 4) infested with *C. felis*-positive *Amblyomma americanum* adult ticks were available from a previously published study [12]. Serial blood samples were collected from these cats at 0, 7, 10, 12, 15 and 19 days post-infestation. Confirmation of *C. felis* infection was carried out by ddPCR as previously described [12].

### 2.3. Clinical Samples (Groups 2 and 3)

Anticoagulated (EDTA) plasma samples (*n* = 46) were obtained from cats presented to veterinary clinics located in Arkansas, Oklahoma and Missouri (USA) from May 2020 to May 2021 with clinical signs suspicious of acute cytauxzoonosis, identified by three (3) or more of the following inclusion criteria: fever > 102.5°F, icterus, depression, anorexia, dyspnea, tachycardia [8]. These samples were classified into two groups: cats confirmed with acute cytauxzoonosis by ddPCR (group 2, *n* = 36) and *C. felis*-negative cats with systemic illness (group 3, *n* = 10) (negative ddPCR and blood smear evaluation). 

### 2.4. Samples from Chronic C. felis Carriers (Group 4)

Anticoagulated (EDTA) plasma samples (*n* = 4 in total) were utilized from three (3) *C. felis* survivor cats housed at Oklahoma State University Animal Resources, and from one (1) cat that presented to OSU-Veterinary Teaching Hospital (OSU-VTH) Small Animal Internal Medicine service for hepatobiliary disease and suspected cytauxzoonosis relapse (incidental finding of intra-erythrocytic piroplasms on serial blood smear evaluations throughout hospitalization). *Cytauxzoon felis* infection was confirmed by ddPCR in all four (4) samples. These samples were classified into group 4. 

### 2.5. C88 Recombinant Protein

Recombinant protein (rProtein) corresponding to an open reading frame encoding a *C. feli*s transmembrane antigen (contig00088:95434-96586(-), hereafter termed c88 rProtein) was synthesized by GenScript Biotech (Piscataway, NJ, USA) according to previously published sequences [11]. The target DNA sequence of c88 rProtein was synthesized and cloned to vector pET30a with His tag for protein expression in *E. coli*. The cells were harvested by centrifugation. Western blot and SDS-PAGE were utilized to determine the protein purity and molecular weight.

### 2.6. Microsphere Immunoassay (MIA)

Evaluation of *C. felis*-specific IgM and IgG antibodies in plasma from cats with experimental *C. felis* infection was performed via microsphere immunoassay (MIA) and conjugation of carboxylated magnetic microspheres (MagPlex® Microspheres, Luminex, Austin, TX, USA) with c88 rProtein according to previously established conjugation protocols [13]. Following conjugation, microsphere concentration was determined by hemocytometer and protein coupling was confirmed via incubation of microspheres with primary antibodies and/or PE-conjugated detection antibodies [13,14]. Successful coupling was defined by a median fluorescence intensity (MFI) of >2000. All samples from experimentally infected *C. felis*-positive cats were diluted 1:50 in assay buffer and then incubated in duplicate with ~2500 conjugated beads per well. All samples were assayed in conjunction with *C. felis*-positive and *C. felis*-negative control samples (outlined above) diluted 1:50 in assay buffer, as well as two diluent control wells per experiment. The MFI was calculated from ≥100 microspheres per analyte per well (Bio-Plex™ Manager 5.0) and then used for data analysis. Mean and standard deviations of MFI of day 0 samples (confirmed negative by *C. felis* ddPCR) were calculated and used as negative controls for MIA analysis. The cut-off for positive detection of anti-c88 IgM and IgG antibodies was defined as mean + standard deviation × 2 of negative controls as previously published [15]. All reagent concentrations, volumes, incubation times, acceptable standard recovery, and data analysis were as previously described [13,14]. 

### 2.7. C. felis IgM/IgG ELISA

An indirect ELISA was designed as previously published [13,16] with modifications to detect *C. felis*-specific IgM and IgG antibodies in plasma samples from cats with acute cytauxzoonosis. Briefly, Corning^®^ 96-well Clear Flat Bottom Polystyrene High Bind Microplates (Corning Inc. Glendale, AZ, USA) were coated with 100 uL/well c88 rProtein in 0.1 M carbonate buffer (IgM: 5 μg/mL; IgG: 2.5 μg/mL) and incubated overnight at 4 °C. After discarding the coating solution, the plate was blocked in 2% bovine serum albumin in TEN buffer (0.1 M Tris-Cl, 0.01 M EDTA, 1 M NaCl) (200 μL/well) for 4 hours at room temperature. The blocking solution was discarded upon completion of blocking. Samples diluted 1:500 in ELISA diluent (4% *v/v* fetal bovine serum, 0.5% *v/v* Triton X-100, 2% bovine serum albumin in TEN buffer), positive control, negative control, or assay buffer (blank) were added (100 μL/well) and incubated at 37 °C for one hour. The plate was washed 5 times with 200 μL/well 0.2% Tween in TEN buffer. HRP conjugated goat anti-feline IgM (Bethyl Laboratories, Inc, Montgomery, TX, USA in-house conjugation performed with LYNX RAPID HRP ANTIBODY CONJUGATION KIT, Bio-Rad Laboratories, Hercules, CA, USA) were diluted 1:20,000 in ELISA diluent and added into wells (100 μL/well). Similarly, HRP conjugated goat anti-feline IgG-Fc fragment (Bethyl, conjugated as above) were utilized at 1:40,000 dilution. The plate was incubated at 37° C for one hour, washed 5 times with 0.2% Tween in TEN buffer, and then 100 μL 1-Step™ Ultra TMB-ELISA Substrate Solution (ThermoFisher™Scientific. Waltham, MA, USA) was added into each well and incubated at room temperature for 10 minutes. Following incubation, 75 μL ELISA Stop Solution (ThermoFisher™Scientific) was added to each well to terminate the reaction. Absorbance was measured immediately at 450 nm by SpectraMax® Plus 384 Microplate Reader (Molecular Devices, LLC. San Jose, CA, USA). The concentration of coating antigen, detection antibody and positive/negative controls were optimized before testing clinical plasma samples. All samples were run in duplicate.

### 2.8. Statistical Analysis

Repeated measures ANOVA was utilized to evaluate differences in anti-c88 IgM and anti-c88 IgG antibodies in plasma of experimentally infected cats over time. Absorbance data from indirect IgM and IgG ELISAs were plotted on Microsoft Excel 16.16.27 software (Redmond, WA, USA) to calculate mean, standard deviation, and percentage of coefficient of variation (% CV) between duplicate samples within the same plate to ensure acceptable intra-plate consistency (% CV < 10%). Each sample was run in duplicate on two separate plates whenever possible to obtain mean absorbance values. The mean absorbance value for each sample was transformed into percent positivity (PP) using a previous published formula [15,17,18] to reduce inter-plate variability. The PP values were imported into GraphPad Prism 8.0 software (La Jolla, California, USA, USA) for receiver operating characteristic (ROC) curve analysis and determination of the assay cut-off value for optimal sensitivity and specificity. The differences in mean PP value between control cats, cats with acute cytauxzoonosis, cats with non-*C. felis* systemic illness and chronic *C. felis* carriers were compared by one-way ANOVA and Dunnett’s multiple comparison test. *p* < 0.05 was considered significant in all analyses.

## 3. Results

### 3.1. Cats with Experimental C. felis Infection Generate IgM and IgG Antibodies against C. felis C88 Recombinant Protein

As shown in Figure 1, Anti-c88 IgM and IgG antibodies were detected by MIA in plasma samples from cats with experimental *C. felis* infection. Anti-c88 IgM antibodies were first detectable at 12 days post-infestation (dpi) (within 24 hours of developing clinical signs) in 75% (3/4) of the experimentally infected cats, and at 15 dpi in 100% (4/4) of cats. Anti-c88 IgM antibodies increased significantly over time (*p* = 0.007), and post hoc analysis revealed that anti-c88 IgM antibodies were significantly elevated over day 0 samples at both 15 dpi (*p* < 0.05) and 19 dpi (*p* < 0.0001). Anti-c88 IgG antibodies were first detectable at 15 dpi (three days after the start of clinical signs) in 75% (3/4) of the cats, and at 19 dpi in 100% (4/4) of cats. Anti-c88 IgG antibodies also increased significantly over time (*p* < 0.0001), and post hoc analysis revealed that anti-c88 IgG antibodies were significantly elevated over day 0 samples at 19 dpi (*p* < 0.0001). These results indicate that although cats generate both IgM and IgG antibodies against *C. felis* during acute infection, anti-*C. felis* IgM antibodies are generated earlier in the course of infection than IgG, and detection of anti-*C. felis* IgM coincides with the onset of clinical signs.

### 3.2. Cats with Acute Cytauxzoonosis Have Significantly Higher Plasma Anti-C88 IgM and IgG Antibodies Compared to Cats in Other Groups 

As outlined in methods, optical density (OD) values of the samples were compared to the OD of positive controls to obtain percent positivity (PP) to allow inter-plate comparisons (see ELISA Data, Appendix A). For both the IgM and IgG ELISAs, mean PP values were compared by one-way ANOVA and revealed statistically significant differences between the four study groups. Post hoc analysis (Dunnett’s multiple comparison test) revealed significant differences (*p* < 0.0001) between healthy control cats (group 1) and cats with acute cytauxzoonosis (group 2) samples in both IgM and IgG ELISAs, as shown in Figure 2. While compared to healthy control cats (group 1), *C. felis* negative cats with systemic illness (group 3) and chronic *C. felis* carriers (group 4) were not significantly different. 

### 3.3. C. felis IgM ELISA Is Highly Specific and Can Successfully Detect Infection in Cats with Acute Cytauxzoonosis

When compared to *C. felis* negative, healthy control cats (group 1) and *C. felis* negative cats with systemic illness (group 3), 34/36 cats with acute cytauxzoonosis (group 2) had high IgM PP values easily distinguished from *C. felis* PCR-negative cats (Figure 3). Percent Positive (PP) values were analyzed by receiver operating characteristic (ROC), and the area under the curve (AUC) was 0.9828 at 95% confidence interval (0.9809–1.000). 

Based on the ROC analysis and AUC results (Figure 4), the *C. felis* IgM ELISA can reliably differentiate cats with and without acute cytauxzoonosis with 100% specificity above 19.85 percent positive (PP). Sample values greater than this cut-off value (34/36 cats in group 2) yields an assay sensitivity of 94.44% and is indicative of a “Strong Positive” result. Although there were no false positive results, infection was not detectable in 2 of the 36 cats with acute cytauxzoonosis at this cut-off value. However, when the cut-off value was lowered to 8.60 percent positive, infection could be detected in these 2 cats with 100% sensitivity and 87.88% specificity. Thus, samples with PP values between 8.60–19.85 are indicative of a “Weak Positive” result. The PP range and interpretation of ELISA results are summarized in Table 1.

### 3.4. Weak Positive IgM ELISA Results Can Be Evaluated in Combination with Other Diagnostics to Reach a Definitive Diagnosis 

Based on the cut-off values proposed in above, two PCR-positive cats with acute cytauxzoonosis (group 2) and two *C. felis* PCR-negative, febrile cats (group 3) had “Weak Positive” IgM ELISA results. Blood smear evaluation of these samples revealed low numbers of intra-erythrocytic piroplasms in both of the group 2 cats (Appendix A) and none in the two group 3 cats (Appendix A), indicating that additional diagnostics such as blood smear evaluation and/or PCR could aid to confirm the diagnosis. 

### 3.5. C. felis PCR Negative Cats Have Lower IgG than Cats with Acute Cytauxzoonosis

As shown in Figure 5, the *C. felis* IgG ELISA in general had excellent performance in differentiating healthy control cats (group 1) and *C. felis* PCR negative, febrile cats (group 3) from cats with acute cytauxzoonosis (group 2). When the cutoff percent positivity was set at 9.90, the IgG ELISA had 100% sensitivity and 96.55% specificity in detecting cats with acute cytauxzoonosis in the current study set (Figure 6). However, one cat (cat 310) in group 3 had a high IgG PP value (48.3) comparable to cats with acute cytauxzoonosis, indicating the potential for previous exposure to *C. felis*. 

### 3.6. Asymptomatic Chronic C. felis Carriers Exhibit High IgG but Variable IgM Antibody Levels 

All *C. felis* chronic carrier cats (4/4, group 4) had higher IgG PP values than the healthy control cats (22/22, group 1) and 9/10 of the *C. felis* PCR negative, febrile cats (group 2). While most of the chronic carriers had low IgM PP values similar to values in group 1 and group 2 cats, one chronic carrier exhibited a high IgM PP value (40.3) (Figure 3). However, this cat was clinically healthy and did not exhibit any clinical signs of cytauxzoonosis.

## 4. Discussion

Because treatment of cytauxzoonosis must be instituted early in the course of disease to be effective, and due to inherent difficulties in confirming *C. felis* infection prior to instituting therapy, we sought to develop the *C. felis* anti-c88 IgM ELISA as a serodiagnostic test to detect *C. felis* infection in cats with clinical signs of acute cytauxzoonosis. Initial results of this study confirmed that both IgM and IgG antibodies against *C. felis* c88 rProtein were detectable and increased over time in experimentally infected cats, justifying the use of this recombinant protein in the development of indirect ELISAs in the second phase of the study. Antibody responses measured by MIA from serial plasma samples from cats with experimental *C. felis* infection also revealed that anti-c88 IgM antibodies were detected earlier than anti-c88 IgG antibodies and coincide with the onset of clinical signs, making the anti-c88 IgM antibody a potential detection marker for early infection.

Overall, the *C. felis* anti-IgM ELISA has excellent performance in diagnosing *C. felis* infection in cats with clinical signs of acute cytauxzoonosis based on the ROC characteristic. With the proposed cut-off value (PP = 19.85) to indicate a “Strong Positive” result, the sensitivity and specificity of the assay is 94.44% and 100%, respectively. These results indicate that the *C. felis* anti-c88 IgM ELISA is highly specific in diagnosing acute cytauxzoonosis therefore can be used as a rule-in test. Even though the sensitivity at this cut-off is excellent, however, there is the slight potential for false negative results. The ROC analysis provides a trade-off between sensitivity and specificity of a given diagnostic test, and based on the context of the data to be analyzed, a cut-off value is set to reach optimal performance of the assay. If we select a cut-off value that carried a higher sensitivity and slightly lower specificity, some cats without acute cytauxzoonosis could be misdiagnosed (false positives) based on ELISA only. Due to the historically guarded prognosis of the disease in some endemic areas of the United States [6,7,8], and considering the relatively serious consequences (euthanasia) for false positive results, we classified the positive ELISA results into “Strong Positive” and “Weak Positive” based on different ranges of cut-off PP values. With the “Strong Positive” cut-off, we maximized the assay specificity (100%) to eliminate possibilities of a false positive result in cats with clinical sings of cytauxzoonosis. With the “Weak Positive” cut-off, we increased the assay sensitivity and lowered the assay specificity, which allowed us to detect all PCR-positive cats with acute *C. felis* infection. 

Within the “Weak Positive” cut-off value range of 8.60–19.85 PP, two *C. felis* PCR-positive cats (group 2) and two *C. felis* negative cats with systemic illness (group 3) were detected (Figure 3), indicating that ELISA results of these cats should be interpreted with caution. Thus, in the event of a “Weak Positive” result, follow-up diagnostics such as blood smear evaluation and/or PCR is warranted to confirm diagnosis. Blood smears were made and evaluated based on identification and quantification of intra-erythrocytic piroplasms and schizont-laden macrophages in all clinical blood samples as previously described [12]. For the two cats with acute cytauxzoonosis and “Weak Positive” IgM ELISA results in group 2 (cats 205 and 209; Appendix A), blood smear evaluation revealed 1-2 piroplasms per 10 high power fields, confirming diagnosis of *C. felis* infection. Cross-reactivity for infection with other piroplasms is highly unlikely since *Cytauxzoon felis* is the only known piroplasm of domestic cats in the United States [19]. The relatively low numbers of piroplasms identified in these cats indicate that they likely presented clinically very early in the course of infection, since intra-erythrocytic piroplasms (merozoites) typically occur during the later course of parasite life cycle in cats [20]. This could also explain why these two samples had “Weak Positive” IgM ELISA results, since based on the antibody responses measured from experimentally infected cats, a patent period exists where infection was not detectable (Figure 1). For the two *C. felis* negative cats with systemic illness and “Weak Positive” IgM ELISA results in group 3 (cats 304 and 308; Appendix A), no piroplasms or schizont-laden macrophages were identified in addition to negative PCR results. Thus, for samples with challenging blood smear evaluation and “Weak Positive” IgM ELISA results, PCR would be necessary as a confirmatory diagnostic test. Due to the turnaround time of currently available *C. felis* PCR assays and the negative outcome of the disease if treatments were not initiated promptly, empirical treatment of anti-protozoal therapy before the PCR result becomes available would be recommended. 

As shown in Figure 2B and Figure 5, cats with acute cytauxzoonosis (group 2) had significantly higher PP values on the anti-c88 IgG ELISA as compared to healthy control cats (group 1). Anti-c88 IgG ELISA PP values were also higher in group 2 compared to *C. felis* PCR negative cats with systemic illness (group 3) and *C. felis* chronic carriers (group 4), although these values were not statistically significant. With the cutoff percent positivity set at 9.90, the IgG ELISA had excellent performance in detecting cats with acute cytauxzoonosis (100% sensitivity and 96.55% specificity). However, one cat in group 3 (cat 310), had a high IgG PP value (48.30). Since this sample was acquired from a cat with suspected acute cytauxzoonosis that was presented to a veterinary clinic in southwestern Missouri, a *C. felis* enzootic area where survivors are not uncommon, it is speculated that this cat might have been previously exposed to cytauxzoonosis, resulting in seroconversion and high circulating IgG concentration. The finding in this cat not only negatively impacted the IgG assay specificity, but also indicated that the IgG ELISA is less than ideal to detect active infection, considering the aim of developing this assay as a point-of-care diagnostic and the fact that the majority of cats tested positive in enzootic areas could be euthanized due to poor prognosis. 

IgM and IgG antibodies were measured by anti-c88 indirect ELISAs in four *C. felis* chronic carriers in the current study set. Differentiating acute infection vs. chronic carrier (or previous *C. felis* exposure) could be valuable in some *C. felis* enzootic areas where survivors are more common, especially for *C. felis* chronic carriers presented for other illnesses. In general, the IgM response is elicited early during the course of infection and wanes over time and could therefore be used as a marker for acute infection. On the other hand, the IgG response is elicited later during infection and could remain measurable in variable lengths of time and could therefore be used as a marker for exposure to a disease. We expected to see *C. felis* chronic carriers to have low IgM antibody levels but high IgG antibody levels, and the reverse for acutely infected cats. Although the *C. felis* chronic carriers had high PP values on the IgG ELISA, they had variable values on the IgM ELISA. Particularly, one cat in group 4 (chronic *C. felis* carriers) had very high IgM PP value (Figure 3), comparable to many cats in group 2 (acute cytauxzoonosis). While this cat was asymptomatic at the time of sample collection, it has historically tested positive on various *C. felis* PCR assays, and its parasite load is known to be high (10,430 copies/reaction) [1,12]. A possible explanation could be an immunological inability to clear the parasite, resulting in continual replication and release of the parasites and intermittent production of IgM antibodies [21,22]. During human malarial infection, the protozoal parasite *Plasmodium vivax* is known to alter the host B-cell profile and therefore result in a persistent malarial-specific IgM response [23]. Thus, it is reasonable to speculate that *C. felis* could elicit a similar response in the host immune system which could account for the high anti-*C. felis* IgM level observed in the current study. 

Most cats with acute cytauxzoonosis (group 2) had detectable IgG antibodies on presentation to the veterinary clinics, making the utility of the anti-c88 IgG ELISA limited in differentiating acute vs. chronic infection. Due to this scenario, we recommend that interpretation of IgM ELISA results should be considered in the proper clinical context such as clinical history, symptoms, or other laboratory findings suspicious of acute cytauxzoonosis. This could be of particular importance in endemic areas where known *C. felis* survivors or cats with incidental finding of previous exposure are present. 

One major limitation of this study is the small sample size, particularly in group 3 and group 4 cats. The types of samples applicable for this assay could be another limitation. Due to sample availability, we were only able to use EDTA-anticoagulated plasma for this study. However, considering the potential for this ELISA to become a bedside test in clinical setting, it is worthwhile to expand the testing to other sample types such as whole blood, heparinized plasma, serum, or citrated plasma for further validation of the assay on these samples. Other than applying this ELISA to more clinical cats and further validating it to the use of other types of sample preparation, it is also valuable to investigate and evaluate the same ELISA design using other *C. felis* putative antigens discovered previously [11] to compare their capabilities in the early detection of feline acute cytauxzoonosis. 

## 5. Conclusions

The *C. felis* anti-c88 IgM ELISA is a highly specific and sensitive test in diagnosing acute cytauxzoonosis in clinical cats presented with comparable history and symptoms. Anti-*C. felis* IgM antibodies are generated earlier in the course of infection than IgG, and detection of anti-*C. felis* IgM coincides with the onset of clinical signs – making it a valuable marker to detect acute *C. felis* infection in symptomatic cats. It is a relatively non-invasive as compared to sedated fine needle aspirations and has the potential to be developed into a rapid bedside screening test with advantageous diagnostic turnaround as compared to PCR.

## Figures and Tables

**Figure 1 pathogens-11-01183-f001:**
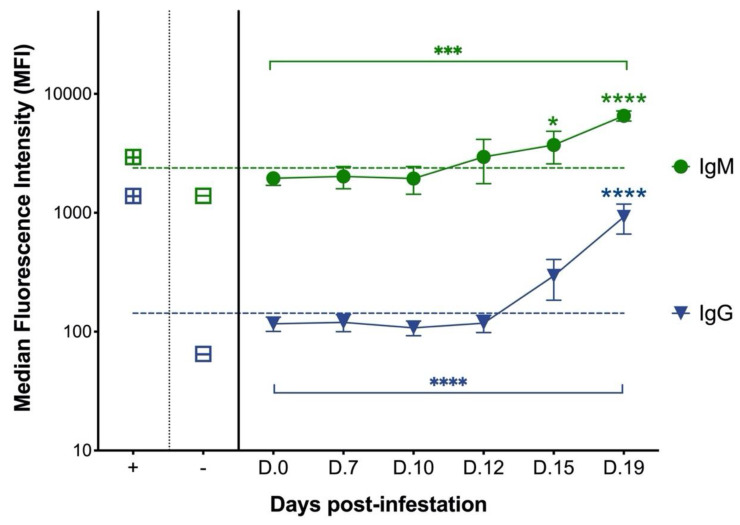
**Anti-c88 IgM and IgG antibodies increase over time in cats with experimental *C. felis* infection.** Plasma concentrations of anti-c88 IgM and IgG antibodies from cats with experimental *C. felis* infection were measured by microsphere immunoassay. Relative plasma concentrations of the antibodies were expressed as median fluorescence intensity. Anti-c88 IgM antibodies were first detected at 12 dpi and increased over time (*p =* 0.0007) and at individual time points compared to day 0 (pre-infection). Anti-c88 IgG antibodies also increased over time and at individual time points but were not detected until 15 dpi. Squares with “+” indicate MFI of positive control plasma. Squares with “-” indicate MFI of negative control samples. Dashed line represents the positive cut-off limit for detection of anti-c88 IgM and IgG antibodies as described in methods. (***** = *p* < 0.05, ******* = *p* < 0.001, ******** = *p* < 0.0001).

**Figure 2 pathogens-11-01183-f002:**
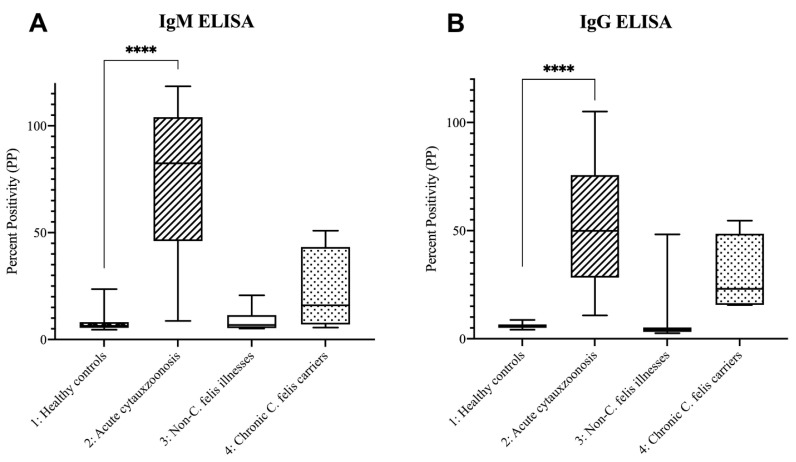
**Cats with acute cytauxzoonosis had higher plasma IgM and IgG antibodies as compared to cats in other groups.** Average (**A**) IgM and (**B**) IgG optical absorbance values (presented as PP) between different groups of cats are shown. For both IgM and IgG ELISAs, only cats with acute cytauxzoonosis (group 2) had significantly higher average PP values (*p* < 0.0001) when compared to healthy control cats (group 1). The average IgM and IgG PP values were not significantly different in cats with non-*C. felis* illness (group 3) or *C. felis* chronic carriers (group 4). (******** = *p* < 0.0001)).

**Figure 3 pathogens-11-01183-f003:**
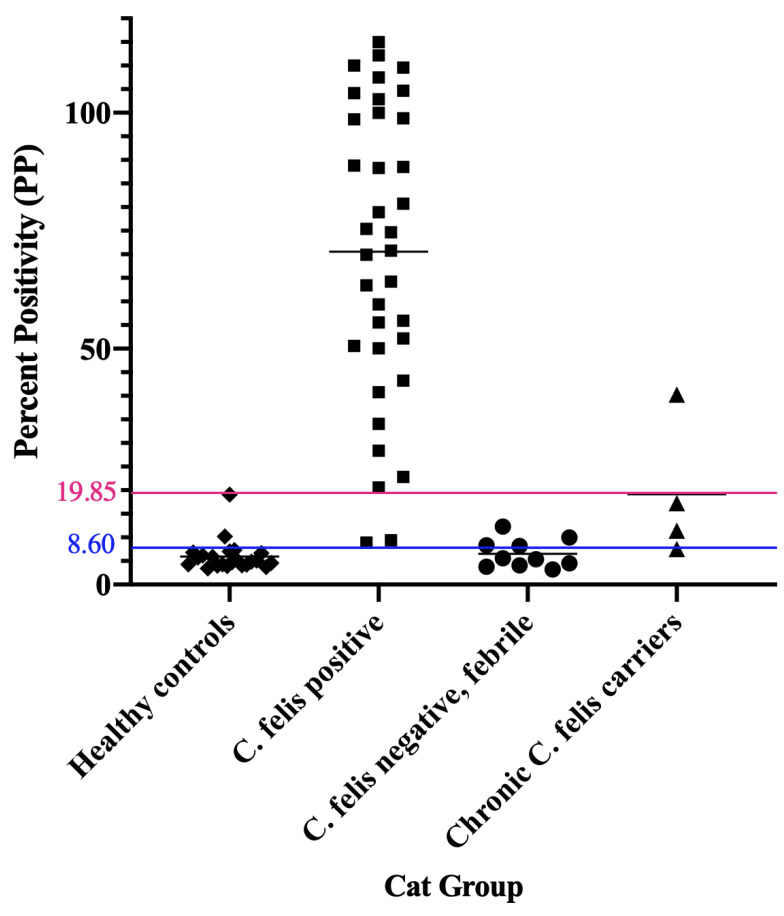
**The anti-c88 IgM ELISA can confidently detect 94.44% of the cats with acute cytauxzoonosis.** When the cut-off PP value was set at 19.85 (pink line), The ELISA can confidently detect infection in 94.44% (34/36) of the cats in group 2. No false positives were noted in either group 1 or group 3 cats. Samples with PP above **19.85** are considered “**Strong Positives**”. When the cut-off PP value was lowered to **8.60** (the blue line), all cats in group 2 could be detected. Samples with PP value between **8.60–19.85** are considered “**Weak Positives**”. Group 1: healthy control cats; Group 2: cats with acute cytauxzoonosis; Group 3: cats with non-*C. felis* illness; Group 4: chronic *C. felis* carriers; Each dot represents a different sample; The thin black line in each group represents the mean PP values within the group; The pink line represents the cut-off PP value with 100% specificity and 94.44% sensitivity. The blue line represents the cut-off PP value with 87.88% specificity and 100% sensitivity.

**Figure 4 pathogens-11-01183-f004:**
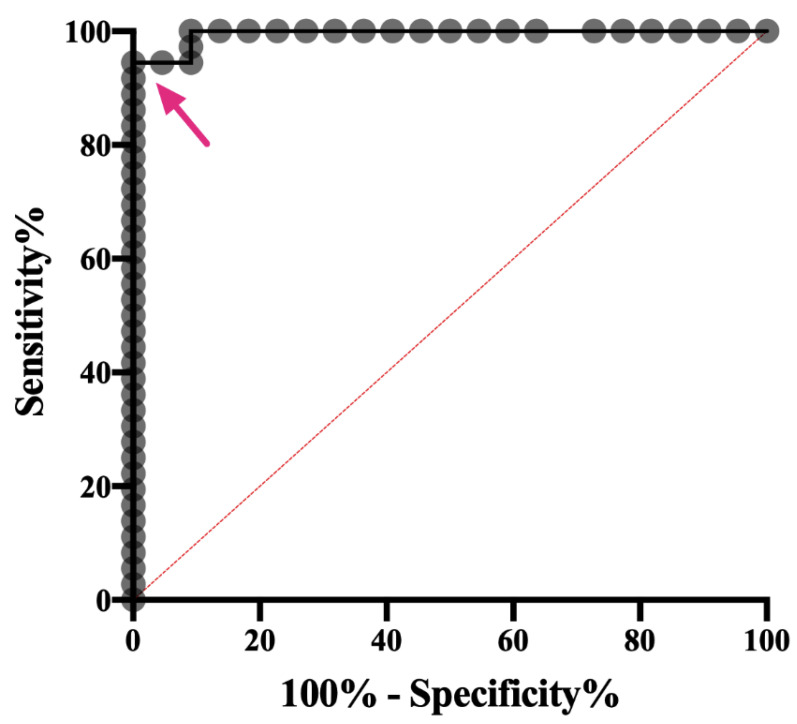
**Receiver operating characteristic curve of the *C. felis* anti-c88 IgM ELISA.** Receiver operating characteristic (ROC) curve demonstrates the performance of the ELISA in differentiating cats with or without acute cytauxzoonosis. The AUC (area under the curve) was 0.9828 at 95% confidence interval (0.9809–1.000). The arrow indicates the cut-off for strong positive results (PP > 19.85) with 100% specificity and 94.44% sensitivity.

**Figure 5 pathogens-11-01183-f005:**
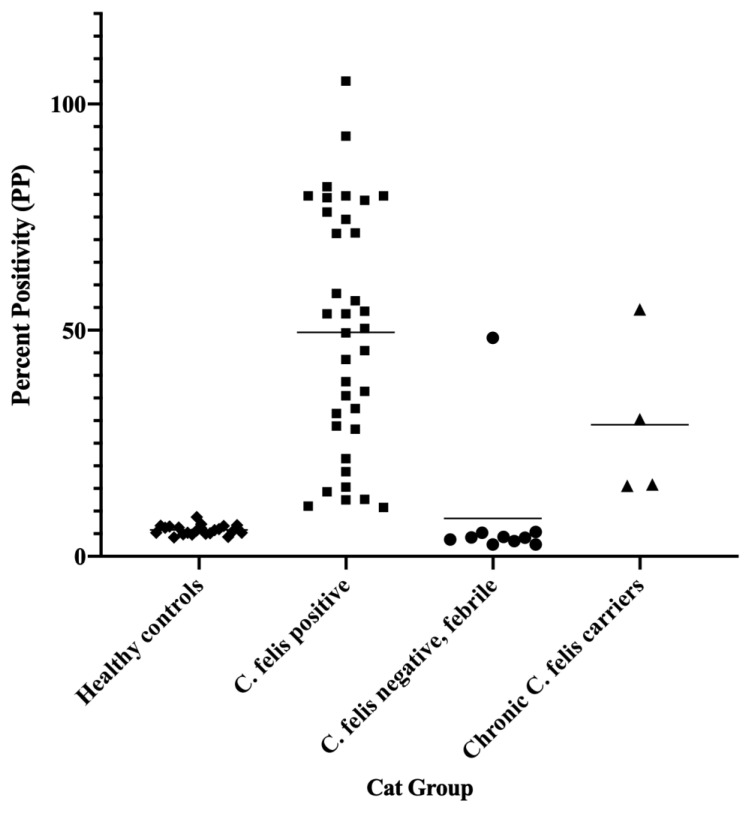
**The majority of cats with acute cytauxzoonosis had higher IgG PP values as compared to cats in other groups.** All healthy control cats (group 1) had distinctively lower PP values compared to *C. felis* PCR positive cats (group 2). Although 9/10 *C. felis* PCR negative, febrile cats (group 3) had distinctively lower PP values compared to *C. felis* PCR positive cats (group 2), one cat in group 3 had significantly high PP value (48.30) that was comparable to the group 2 cats, suggesting the potential for previous exposure to *C. felis*. All *C. felis* chronic carriers (group 4) had high IgG PP values in the absence of clinical signs, indicating a chronic immune response to previous *C. felis* infection.

**Figure 6 pathogens-11-01183-f006:**
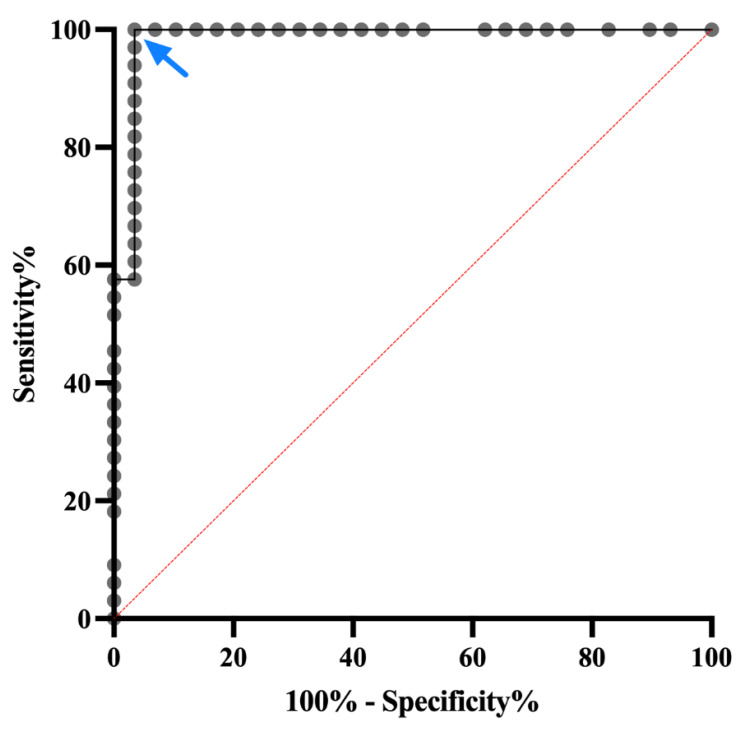
**Receiver operating characteristic curve of the *C. felis* anti-c88 IgG ELISA.** Receiver operating characteristic (ROC) curve demonstrates the performance of the ELISA in differentiating cats with or without acute cytauxzoonosis. The AUC (area under the curve) was 0.9854 at 95% confidence interval (0.9560–1.000). The arrow indicates the cut-off for positive results (**PP > 9.90**) with 96.55% specificity and 100% sensitivity.

**Table 1 pathogens-11-01183-t001:** Interpretation of the *C. felis* IgM ELISA.

PP	Specificity/Sensitivity	Interpretation	Comments
>19.85	100%/94.44%	Strong Positive	Cytauxzoonosis is confirmed.
8.60–19.85	87.88%/100%	Weak Positive	Possible cytauxzoonosis if clinical history is suggestive. Additional diagnostics are recommended (i.e., Blood smear evaluation and/or PCR). Empirical therapy should be initiated while awaiting other diagnostic results.
<8.60	N/A	Negative	Cytauxzoonosis is unlikely.

PP = percent positivity; N/A = not applicable.

## Data Availability

The datasets generated during and/or analyzed during the current study can be find in the main text and the Appendix A (ELISA Data).

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
