# Peer review of "A Serodiagnostic IgM ELISA to Detect Acute Cytauxzoonosis"

_pathogens, 2022, doi:10.3390/pathogens11101183_

Round 1
Reviewer 1 Report
The manuscript entitled “A serodiagnostic IgM ELISA to detect acute cytauxzoonosis” describes the development of a IgM based ELISA for the detection of active infection of Cytauxzoon felis in cats. The manuscript is written in a very nice manner. The topic is very much important. However, some important problems are there which may be considered while revising the manuscript.
1. The disease may cause mortality up to 100 %. There is no vaccine or no successful treatment. Then what is the use of early diagnosis of the diseases? Please explain this in the introduction.
2. The manuscript is silent on the cross-reactivity of the newly developed IgM-based ELISA with other infections caused by the protozoans or any other parasitic organisms.
3. Is it possible that the weekly positive samples are positive for any other protozoan disease?
Minor points
1. Many sentences begin with abbreviations. For eg: C. felis, SDS-PAGE, EDTA, PP values etc. It is better that abbreviations are not used at the start of a sentence.
2. µl, ml, etc are written in many places. It is better to write them as µL, mL, etc
3. Many sentences start with numbers (100µl). This is not appropriate.
4. Table heading should not be ending with a full stop.
5. It is better if you do not write the single-digit number in the text. For eg; better to write five instead of 5. (lines 113, 125, 131, 132, 136, and many places)
Author Response
Response to Reviewers
Reviewer 1
The manuscript entitled “A serodiagnostic IgM ELISA to detect acute cytauxzoonosis” describes the development of a IgM based ELISA for the detection of active infection of Cytauxzoon felis in cats. The manuscript is written in a very nice manner. The topic is very much important. However, some important problems are there which may be considered while revising the manuscript.
We sincerely thank the reviewer for these kind comments, and for the time spent reviewing our manuscript. We have compiled a detailed response to the reviewer’s concerns and suggestions below.
- The disease may cause mortality up to 100 %. There is no vaccine or no successful treatment. Then what is the use of early diagnosis of the diseases? Please explain this in the introduction.
In the Introduction, the text says specifically that “mortality in these regions can reach 100% without treatment”. We then go on to discuss that “Current therapies (atovaquone + azithromycin) have been shown to increase survival up to 60% in cats with cytauxzoonosis, but are hindered by a narrow window in which treatment must be instituted early in the course of disease in order to be effective.” We have revised the last sentence of the Introduction to reiterate the critical need for this diagnostic test: “Because of the rapid disease progression and severe clinical consequence, early detection and diagnosis of cytauxzoonosis is crucial for effective treatment.”
- The manuscript is silent on the cross-reactivity of the newly developed IgM-based ELISA with other infections caused by the protozoans or any other parasitic organisms.
The purpose of this diagnostic assay is to rule in C. felis infection in domestic cats exhibiting clinical signs of cytauxzoonosis. Unfortunately, testing for cross-reactivity is not possible at this time due to lack of samples and/or evidence of other piroplasms causing natural infection in domestic cats of North America. Nevertheless, we have added text to the Discussion to address the subject of cross-reactivity with Weak Positive results in the manuscript.
Cytauxzoon felis is the only known piroplasm of domestic cats in the United States (Reichard et al. 2021). Although an unidentified Babesia was found in 32 of 41 cougars from Florida (Yabsley et al. 2006), five of these same 41 cougars were infected with Cytauxzoon felis, two of the infected cougars had both C. felis and the unidentified Babesia sp., and this unidentified Babesia species from cougars in Florida has not been identified from domestic cats. Shock et al (2013) found an unidentified Babesia sp. from a single bobcat in Georgia, but this unidentified Babesia sp. from this 1 bobcat has not been reported since.
Penzhorn and Oosthuizen 2020 published a review of Babesia sp. that have been reported from domestic cats, but they include no reports of Babesia spp. from domestic cats in North America. Alvarado-Rybak et al. 2016 reviewed Babesia and Theileria species in wild carnivores, but the only report they include is the Yabsley et al. 2006 paper on Babesia sp. and C. felis in cougars from Florida referenced above. Since there are no samples from confirmed cases of Babesia in domestic cats in North America, we can’t test for cross-reactivity against something that is not there.
Three species of Cytauxzoon, C. europaeus, C. ontrantorum, and C. banethi have recently been described from cats in Europe (Panait et al. 2021). These species are genetically distinct from C. felis and have only been reported in felids in Europe. A fourth Cytauxzoon species, C. manul was reported in Pallas cats imported from Mongolia to a private zoological collection in Oklahoma (Reichard et al. 2005), but this piroplasm has not been reported since in North America. While cross-reactivity with other Cytauxzoon species would be valuable, it is beyond the scope of this manuscript, which serves as a diagnostic tool to diagnose C. felis infection in North America.
References:
- Alvarado-Rybak, M.; Solano-Gallego, L.; Millán, J. A review of piroplasmid infections in wild carnivores worldwide: Importance for domestic animal health and wildlife conservation. Parasit. Vectors 2016, 9, 538
- Panait, LC et al. Three new species of Cytauxzoon in European wild felids. Veterinary Parasitology, 2021, 290, 109344
- Penzhorn, B.L.; Oosthuizen, M.C. Babesia species of domestic cats: Molecular characterization has opened Pandora’s Box. Front. Vet. Sci. 2020, 7, 134.
- Reichard, MV, TL Sanders, P Weerarathne, JH Meinkoth, CA Miller, RC Scimeca, C Almazán. Cytauxzoonosis in North America. Pathogens 2021, 10, 1170
- Reichard MV, RA Van Den Bussche, JH Meinkoth, JP Hoover, AA Kocan. A new species of Cytauxzoon from Pallas’ cats caught in Mongolia and comments on the systematics and taxonomy of piroplasmids. Journal of Parasitology 2005, 91, 420-426
- Shock, BC, JM Lockhart, AJ Birkenheuer, MJ Yabsley. Detection of a Babesia Species in a Bobcat from Georgia. Southeastern Naturalist 2013, 12, 243-247
- Yabsley, M.J.; Murphy, S.M.; Cunningham, M.W. Molecular detection and characterization of Cytauxzoon felis and a Babesia species in cougars from Florida. J. Wildl. Dis. 2006, 42, 366–374.
- Is it possible that the weekly positive samples are positive for any other protozoan disease?
Please see response to #2. Regardless, in the case of a weak positive, confirmatory testing by blood smear or PCR therapy would be recommended per the guidelines reported herein. Since C. felis is the only known piroplasm of domestic cats in the United States, detection by this method would be confirmatory, as would a PCR positive result (gold-standard).
Minor points
- Many sentences begin with abbreviations. For eg: C. felis, SDS-PAGE, EDTA, PP values etc. It is better that abbreviations are not used at the start of a sentence.
Thank you for this suggestion, the manuscript has been revised throughout.
- µl, ml, etc are written in many places. It is better to write them as µL, mL, etc
Thank you for this comment, the manuscript has been edited throughout to correct this.
- Many sentences start with numbers (100µl). This is not appropriate.
Thank you, the manuscript has been revised to correct this.
- Table heading should not be ending with a full stop.
Thanks for this suggestion, the full stop in the table heading has been removed.
- It is better if you do not write the single-digit number in the text. For eg; better to write five instead of 5. (lines 113, 125, 131, 132, 136, and many places)
This has been addressed, thank you

Reviewer 2 Report
In a original research article entitled: "Serodiagnostic IgM ELISA for detection of acute cytauxzoonosis," the authors present valuable results on the possibility of using the recombinant c88 transmembrane protein of C. felis to develop an ELISA assay suitable to detect acute cytauxzoonosis with high specificity and sensitivity in clinical samples from domestic cats.
The manuscript is clearly written and the results are presented in an appropriate manner. I believe that the manuscript is adequate for publication in present form. I hope that a point-of-care test based on c88 protein of C. felis will be soon available as comercial rapid test in clinical practice.
Author Response
We sincerely thank the reviewer for these kind comments, and for the time spent reviewing our manuscript!
Reviewer 3 Report
Thank you for this very complete and well documented article. I think this article would gain in clarity and understanding for the reader by simplifying the result part and refocusing on the main message of the article. My comments are in the PDF.

Author Response
Response to Reviewers
Reviewer 3
Thank you for this very complete and well documented article. I think this article would gain in clarity and understanding for the reader by simplifying the result part and refocusing on the main message of the article. My comments are in the PDF.
We sincerely thank the reviewer for these kind comments, and for the time spent reviewing our manuscript. We have compiled a detailed response to the reviewer’s concerns and suggestions below.
Objectives of the study should be placed at the end of the discussion
We suspect that the reviewer meant that the objectives should go at the end of the Introduction. If so, we have strategically placed the objectives as close to the end of the Introduction as possible while still maintaining the intended rationale. We wish to convey how the objectives of the study drove the development of this diagnostic assay, and wish to leave the objectives in their current location.
How was the N chosen for each group? Was it what was available or was a sample size calculation done?
The N for each group was chosen solely on the availability of samples. We included as many as possible.
I think that 6 figures is too much for this part of the results, we get lost in the results and it is difficult to know what are the main results of the study.
We thank the reviewer for this suggestion. We feel the inclusion of each figure is necessary to the overall argument of the results, but if the editors feel otherwise, we can move 1-2 of the figures to supplementary materials.

Round 2
Reviewer 3 Report
thank yo for the update